# Enriched microbial consortia from natural environments reveal core groups of microbial taxa able to degrade terephthalate and terphthalamide

**Laura G. Schaerer** **[1], Sulihat Aloba[2], Emily Wood[1], Allison M. Olson[1], Isabel B. Valencia[1], Rebecca G. Ong[2], Stephen M. Techtmann[1] \***

**1** Department of Biological Sciences, Michigan Technological University, Houghton, Michigan, United States of America, **2** Department of Chemical Engineering, Michigan Technological University, Houghton, Michigan, United States of America

\* smtechtm@mtu.edu

## Abstract

Millions of tons of polyethylene terephthalate (PET) are produced each year, however only ~30% of PET is currently recycled in the United States. Improvement of PET recycling and upcycling practices is an area of ongoing research. One method for PET upcycling is chemical depolymerization (through hydrolysis or aminolysis) into aromatic monomers and subsequent biodegradation. Hydrolysis depolymerizes PET into terephthalate, while aminolysis yields terephthalamide. Aminolysis, which is catalyzed with strong bases, yields products with high osmolality, which is inhibitory to optimal microbial growth. Additionally, terephthalamide, may be antimicrobial and its biodegradability is presently unknown. In this study, microbial communities were enriched from sediments collected from five unique environments to degrade either terephthalate or terephthalamide by performing biweekly transfers to fresh media and substrate. 16S rRNA sequencing was used to identify the dominant taxa in the enrichment cultures which may have terephthalate or terephthalamide-degrading metabolisms and compare them to the control enrichments. The goals of this study are to evaluate (1) how widespread terephthalate and terephthalamide degrading metabolisms are in natural environments, and (2) determine whether terephthalamide is biodegradable and identify microorganisms able to degrade it. The results presented here show that known contaminant-degrading genera were present in all the enriched microbial communities. Additionally, results show that terephthalamide (previously thought to be antimicrobial) was biodegraded by these enriched communities, suggesting that aminolysis may be a viable method for paired chemical and biological upcycling of PET.

## Introduction

Polyethylene terephthalate (PET) is a plastic polymer that is made of repeating building blocks of the aromatic monomer terephthalic acid and ethylene glycol [1] and is widely used for

**Data Availability Statement:** The sequencing data files are available under Bioproject PRJNA1003086.

Code is available at https://github.com/lgschaer/TA_TPA_Enrichment.

**Funding:** This work was supported by the Defense Advanced Research Projects Agency ReSource program cooperative agreement HR00112020033. The views, opinions, and/or findings expressed are those of the author and should not be interpreted as representing the official views or policies of the Department of Defense or the U.S. Government. This work was also supported by the Merck Future Insight Prize.

**Competing interests:** The authors declared that no competing interests exits.

many purposes worldwide, primarily food packaging and apparel [2, 3]. In 2014, 41.56 million metric tons of PET was produced globally; this number is expected to continue to grow exponentially in coming years [4]. Presently, over 70% of manufactured plastic ends up in a landfill or in the ocean since current methods for plastic recycling are time consuming and expensive [5, 6]. While the extent of PET recycling is increasing, only about 29% of PET is recycled in the United States [7]. This accumulation of plastic in the environment is in part due to limitations in the current methods for plastic recycling that are time consuming, expensive, and lead to down-cycling of the material [5, 6] There is a need for alternative methods for recycling and upcycling of PET wastes. In recent years, progress has been made toward using microorganisms to assist in valorization of waste products [8], closed-loop plastic recycling [9], and upcycling plastic into value-added materials [10–12].

Chemical depolymerization of PET through chemical reactions such as hydrolysis and aminolysis produces monomers that are more biologically accessible than plastic polymers. Chemical depolymerization could be integrated into a variety of biological upcycling systems to more efficiently process plastic waste [11]. Hydrolysis produces terephthalate monomers and ethylene glycol [13], and aminolysis produces terephthalamide monomers and ethylene glycol [14]. Terephthalate and terephthalamide are both aromatic monomers. Terephthalate has two carboxylic acid functional groups on either side of an aromatic ring, while terephthalamide has two amide functional groups on either side of an aromatic ring. Aminolysis, unlike hydrolysis, results in a nitrogen-containing product with lower osmolality. Since solutions with high osmolality may inhibit microbial growth [15], and nitrogen is required for microbial growth, use of aminolysis for PET upcycling could streamline the current practice of using hydrolysis. Deconstruction of PET using ammonium hydroxide will result in a mixture of terephthalate, terephthalate monoamide (one amide functional group and one carboxylic acid functional group), terephthalamide, and ethylene glycol. Terephthalate, produced by hydrolysis, biodegradation is already known to be catalyzed by diverse bacteria and has been described in depth in the literature [10, 16, 17]. However, it is currently unknown whether terephthalamide, produced by aminolysis, is biodegradable. If terephthalamide degrading bacteria can be identified, aminolysis may be a practical means for chemically depolymerizing PET and optimizing current bioprocessing methods.

It is currently unknown if microbial metabolisms exist to degrade terephthalamide, which is essential if aminolysis is to be used for PET depolymerization. However, studies have identified organisms able to degrade similar compounds including nitrogen-containing aromatics. Terephthalamide may have antimicrobial properties, and some studies have shown that it is a precursor to effective antimicrobials [18–20]. A fungus, *Aspergillus flavus* was shown to be able to degrade poly *p*-phenylene terephthalamide fibers [21] which consists of amide-linked terephthalic acid and p-phenylene diamine monomers. A few previous studies have identified microorganisms which degrade aromatic nitrogen-containing compounds including *Arthrobacter*, *Pseudomonas*, and *Rhodococcus* [22, 23], some of the same genera which are known to degrade terephthalate. Known terephthalate-degrading bacteria include members from the genera *Rhodococcus*, *Comamonas*, *Delftia* and *Pseudomonas* [24, 25]. These studies suggest that some microorganisms may be able to at least tolerate terephthalamide despite its antimicrobial effects and diverse organisms have the ability to degrade aromatic amides [26]. Considering these previous studies, the existence of naturally occurring microbial metabolisms to degrade terephthalamide is likely.

Recent literature has suggested that the natural environment may be a reservoir of novel microbial metabolisms, which may have useful applications in biotechnological systems[10, 11, 27]. Bioprospecting for organisms that perform biotechnologically relevant chemical reactions has proven fruitful for identification of development of bioindustrial processes. Many

naturally occurring microorganisms have been observed breaking down plastics in a variety of environments (marine [28], soil [29], landfills [30] and sewage [31]). Microorganisms from natural environments with the ability to break down PET [32] and terephthalate [33, 34], have been successfully grown in laboratory cultures and isolated. PET can be naturally deconstructed into aromatic monomers by enzymes called PETases [3, 35] which yields terephthalate. Terephthalate degrading bacteria have been isolated from numerous sources including soil [33], household compost [34], and wastewater [36, 37].

In this experiment, geographically widespread environments were investigated to find microbes that degrade terephthalate and terephthalamide. To our knowledge, this will be the first study showing biodegradation of terephthalamide. To test the hypothesis that terephthalate and terephthalamide degrading microorganisms will be present in a variety of sediments collected from five distinct locations and environments, terephthalate and terephthalamide degrading microbial communities will be enriched by culturing sediment samples with these monomers as the sole carbon source and transferring the cultures several times. To test a secondary hypothesis that terephthalamide will be degraded by the bacteria enriched in the terephthalamide-enriched treatments, the amounts of substrate degraded will be quantified with high performance liquid chromatography.

## Material and methods

### Enrichment of aromatic degrading consortia

Soil and sediment samples were collected from five environments: vermicompost [38] (Calumet, Michigan, USA; coordinates 47.211, -88.553), freshwater lake sediment (Lake Superior, Bete Grise, Michigan, USA; coordinates 47.3723, -87.9529), freshwater shoreline sediment (Straits of Mackinac, Michigan, USA; coordinates 46.0271, -84.5917), central Pennsylvania stream sediment (Fall Run, Pennsylvania, USA; coordinates 41.5468, -76.7726), and sediment from a hydrocarbon seep (Caspian Sea; coordinates 39.7455, 50.4806). No permits were required for sample collection from these sites. Samples were stored at 4 ˚C until used to inoculate enrichment cultures. For each enrichment, 0.5 g of surface soil or surface sediment was used to inoculate 50 mL of Bushnell Haas media containing: magnesium sulfate (0.2 g/L), anhydrous calcium chloride (0.02 g/L), potassium dihydrogen phosphate (1 g/L), dipotassium hydrogen phosphate (1 g/L), ammonium nitrate (1 g/L), and ferric chloride (0.05 g/L). Enrichment cultures were amended with either 10 g/L of disodium terephthalate, 5 g/L of terephthalamide or no added plastic substrate (control). A lower concentration of terephthalamide was used due to potential antimicrobial effects. For each inoculum and substrate, three replicate enrichment cultures were set up in 125 mL Erlenmeyer flasks and incubated at 25 ˚C on stir plates at 130 rpm. Every two weeks (14 days), 5 mL of each culture was transferred to fresh media containing either 0.5 g of disodium terephthalate, 0.25 g terephthalamide or no added carbon source (control). Three aliquots of 1.8 mL were taken from each culture for HPLC analysis and frozen at -20 ˚C immediately. The remaining culture was divided into two 50 mL centrifuge tubes (20 mL ± 2 mL each) and centrifuged at 10,000xg for 20 minutes, yielding two cell pellets for each culture. The supernatant was discarded and DNA was extracted from one pellet while the other was frozen at -80 ˚C as an archive. Cultures were transferred this way a total of four times, sampling was repeated for each iteration.

### 16S rRNA sequencing

DNA was extracted from each cell pellet using MP Biomedicals Fast Soil DNA extraction kit. DNA was sent to Michigan State University Genomics Core for amplicon sequencing of the

V4 region of the 16S rRNA gene. Primers 515F and 806R were used [39]. Sequencing was performed on an Illumina Miseq, producing 250 bp paired-end reads.

## 16S rRNA data analysis

Sequence analysis was performed in R [40], ver. 4.1.3. The DADA2 (divisive amplicon denoising algorithm) package in R [41] was used to trim primers, overlap paired-end reads, quality filter, remove the internal standard (phiX), and infer amplicon sequence variants (ASVs). Using DADA2, denoised reads were merged, bimeric reads were removed, and taxonomy of ASVs was assigned using the Silva database v138. The complete pipeline for the analysis is available at https://github.com/lgschaer/TA_TPA_Enrichment. Quality control blanks (PCR controls and DNA extraction blanks) were removed from the computational analyses. Diversity analysis was performed with the phyloseq package in R [42]. Mitochondrial DNA and chloroplasts were removed, and sequences were rarefed using the *rarefy even depth* function in phyloseq to a minimum sequence sample size of 1053 reads. On the rarefied table, alpha diversity was calculated with the *estimate richness* function in phyloseq using the metrics Shannon and Observed ASVs. A Kruskal–Wallis test (FSA package [43]) was used to determine if there was a statistically significant difference between sample groups. Kruskal–Wallis was chosen because it is compatible with unbalanced and non-normally distributed datasets. Base R [40] was used to perform a Dunn test (post-hoc) to identify significant differences between pairwise comparisons of sample groups. Comparisons of microbial community composition between the different enrichments were performed using the R package phyloseq. A PCoA plot was constructed from a unifrac distance matrix (constructed using the APE package in R [44]) and was used to visualize differences in microbial community composition. A permutational multivariate analysis of variance (PERMANOVA) was performed using the adonis function in the vegan package [45] to identify statistically significant differences between pairwise comparisons of sample types. Differential abundance analysis was performed using the DESeq2 package in R on the unrarefied data [46]. To explore the predicted genomic potential of the organisms enriched by these experiemnts, we used PICRUSt2 [47], which uses 16S rRNA taxonomic assignments and publicly available genome databases to predict genome content of microorganisms. The rarefied ASV table was input into PICRUSt2 and the program was run using default parameters. Tidyverse packages dplyr and ggplot2 were used to filter, summarize, and visualize the results.

## Substrate quantification

The terephthalamide substrate remaining in the media was extracted and quantified for one replicate sample from each environment and transfer. Terephthalamide was extracted from cultures by adding 1 mL of culture to 13 mL of dimethylformamide. The samples were mixed two times for 20 seconds each. The samples were incubated at room temperature for several days, mixing every 2–3 days until solid terephthalamide was no longer visible. Samples were syringe filtered with PTFE 0.22 μ filters. The samples were then quantified using high performance liquid chromatography. Blanks were extracted from subsamples collected from well-mixed flasks with 50 mL Bushnell Haas media amended with 0.25 g of terephthalamide. These blanks were used to test extraction efficiency.

Terephthalic acid was purchased from Millipore-Sigma and terephthalamide was purchased from TCI Chemicals. 1 mg/mL standard solutions of terephthalic acid and terephthalamide were prepared in N, N-dimethylformamide (DMF). The stock solutions were diluted with DMF to give five different concentrations for calibration curves, and filtered through a

0.22 μm polytetrafluoroethylene (PTFE) filter. The calibration curves were obtained by plotting the peak area of each standard to their known concentrations.

Substrate quantification was performed according to the protocols described in Schaerer et al (2023) [11]. Briefly, the analysis was performed with an Agilent 1200 liquid chromatography system equipped with a G1311A quaternary pump, G1322A degasser, G1329A autosampler, G1315B diode-array detector, and 61316A temperature column controller. Separations were performed on Waters μBondapak C18 column (3.9 mm Å~ 300 mm, 10 μm). The mobile phase consisted of 0.2% formic acid-water solution (A) and 0.1% formic acid-acetonitrile solution (B) using a gradient elution: 0–5 min, 20–30% B, 5–10 min, 30–40% B, 10–15 min, 40–50% B, 15–20 min, 50–60% B, 20–25 min, 60–70% B, 25–35 min, 70–90% B, and 35–45 min, 90–20% B. The analysis was carried out at a flow rate of 0.4 mL/min and an injection volume of 20 μL over 45 min. Detection was carried out using a diode-array detector at 300 nm for terephthalic acid and 275 nm for terephthalamide due to their optimum linear standard curves at those wavelengths.

## Results

### The enrichment process selected for a subset of the starting microbial community

Alpha diversity showed that, on average, the inoculum for each environment was the most diverse sample type, followed by control samples, terephthalamide enrichments, and finally the terephthalate enrichments; this pattern held true for both Observed ASVs and Shannon diversity. Of the five environments sampled, only inoculum samples from the compost and the stream sediment sequenced with sufficient reads to be used in computational analysis of sequences (S1 Fig, S1 Table in S1 File). The pattern of alpha diversity shows decreasing diversity with each transfer showing that the enrichment process selected for a small subset of the starting community, except for the Pennsylvania stream sediment in the TPA treatment, alpha diversity increased from Transfer 1 (Observed 40.3 ± 7.77, Shannon 2.2 ± 0.43) compared to Transfer 4 (Observed 65.3 ± 40.1, Shannon 2.5 ± 0.46) (S3 Fig in S1 File) (Fig 1). Similar trends are visible with Shannon diversity, although the median Shannon diversity for the terephthalate treatment had a broader range of values over the course of the experiment (S2 Fig in S1 File). A Kruskall Wallis test showed that there was a statistically significant difference in alpha diversity between different transfers for both Shannon and Observed diversity metrics (S2 Table in S1 File). A Dunn post hoc test was performed to determine whether there was a statistically significant difference in alpha diversity between transfers (i.e. first and second (T1—T2), first and fourth (T1—T4), etc.) (S3 Table in S1 File). The most statistically significant difference was between the first and third transfers (T1—T3) first and fourth transfers (T1—T4), demonstrating that the alpha diversity of the enrichments decreased considerably during the experiment, and that the diversity was largely stable between the third and fourth transfers (T3—T4).

### Microbial community composition of terephthalate and terephthalamide treatments are distinct

Principal Coordinates Analysis (PCoA) was done using a unifrac distance matrix. This showed that the samples clustered loosely by substrate: the control enrichments clustered in the upper left of the figure, there were two clusters of terephthalate samples (one near the upper right and the other on the bottom left of the figure), and the terephthalamide samples cluster in the upper half of the figure, overlapping with both the control samples and the upper right

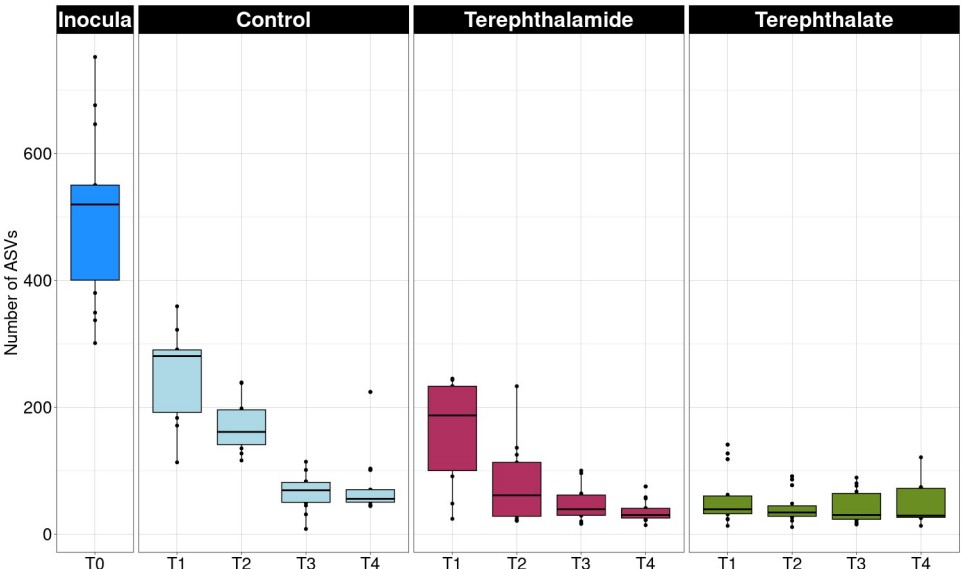

**Fig 1. Observed richness for each substrate over the course of four transfers performed bi-weekly.** The overall trend shows decreasing median diversity with each transfer for each treatment. Each box represents 15 samples (5 environments x 3 replicates).

terephthalate samples (Fig 2A). A pairwise permutational analysis of variance (PERMA-NOVA) was performed to determine if there was a statistically significant difference in microbial community composition between substrate types. The results showed a statistically significant difference between all pairwise comparisons of control, terephthalamide, and

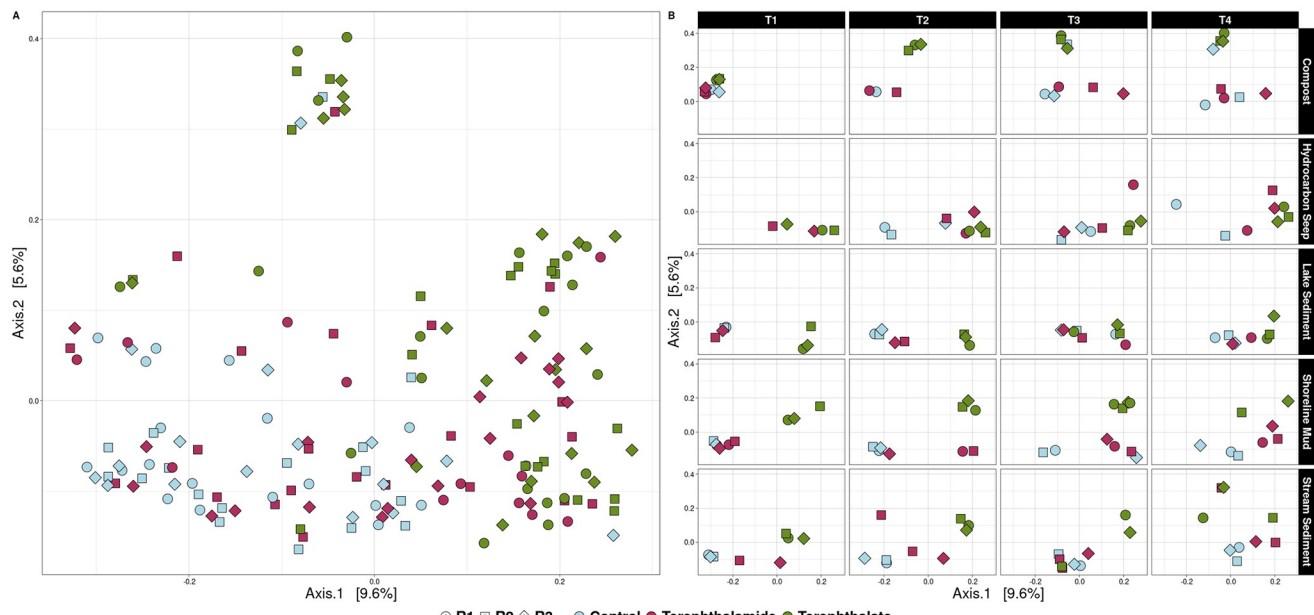

**Fig 2. Unifrac Principal Coordinates Analysis of all samples.** (**A**) All samples from all time points and all sediments sources are displayed on a single panel to visualize larger trends in clustering, (**B**) Figure is shown divided into separate panels based on inoculum source and transfer number. Both panels show clustering based on Substrate (represented by color). Replicates are separate microcosm lines.

terephthalate samples (p-values < 0.01, S4 Table in S1 File). Separating the figure by inoculum source and transfer number confirmed that the samples do cluster primarily by substrate type, most often with the terephthalate samples clustering farthest from the control samples and the terephthalamide samples clustering somewhere in between the control samples and terephthalate samples (Fig 2B). A PERMANOVA was performed to look for differences between the microbial composition of pairwise comparisons of each transfer, grouping all environments and substrate treatments together. The results showed that across all treatments there was a statistically significant difference in microbial community composition between transfer 1 and transfer 3 as well as transfer 1 and transfer 4; the other comparisons were non-significant (S5 Table in S1 File). These results taken together show that microbial community composition was affected by substrate type as well as time (transfer).

## Enriched cultures are dominated by a few taxonomic groups

Over the course of the experiment, diversity decreased, and the cultures became dominated by only a few phyla, specifically, Actinobacteria, Bacteroidota, Firmicutes, and Proteobacteria (S4 Fig in S1 File). The final communities obtained after transfer 4 were dominated by the classes Actinobacteria, Bacteroidia, Bacilli, Alphaproteobacteria, and Gammaproteobacteria (Fig 3).

To investigate taxa that may be involved in the degradation of terephthalate and terephthalamide, the data set was filtered to only include genera that were present at greater than 10% relative abundance in at least one sample of the fourth transfer. The relative abundances of these highly abundant taxa were plotted to identify if any were more prevalent in one treatment compared to the others (Fig 4). Several taxa were much more abundant in the cultures with the added aromatic substrates (terephthalate and terephthalamide) compared to the controls, particularly *Bordetella*, *Micromonospora*, and *Pseudoxanthomonas* (Fig 4). Some taxa were also much more abundant in cultures grown in one substrate or the other. In samples grown on terephthalate, *Flavobacterium* (26% ± 14%), *Hydrogenophaga* (6% ± 8%), and *Rhodococcus* (25% ± 30%) were highly abundant relative to the control samples and the

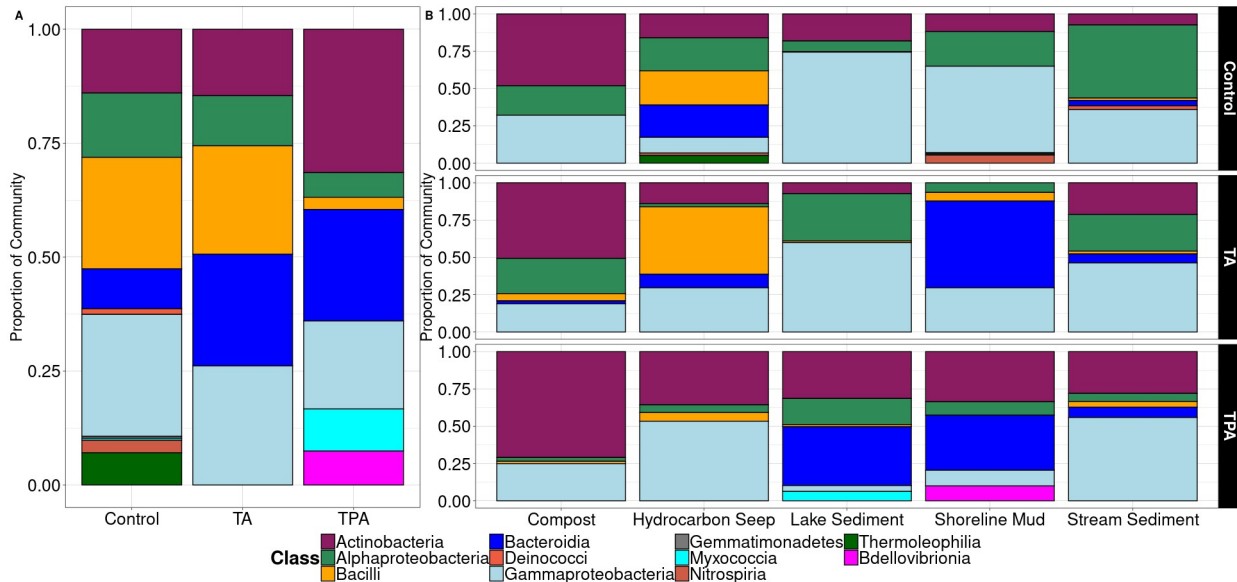

**Fig 3. Taxa plot showing class level classification for the fourth (final) transfer for each substrate.** Bars show the mean relative abundance of each taxon in three replicates across all five inoculum sources. A. Mean relative abundance of each taxon across the five environments. B. Mean relative abundance of each taxon across all replicates for each environment.

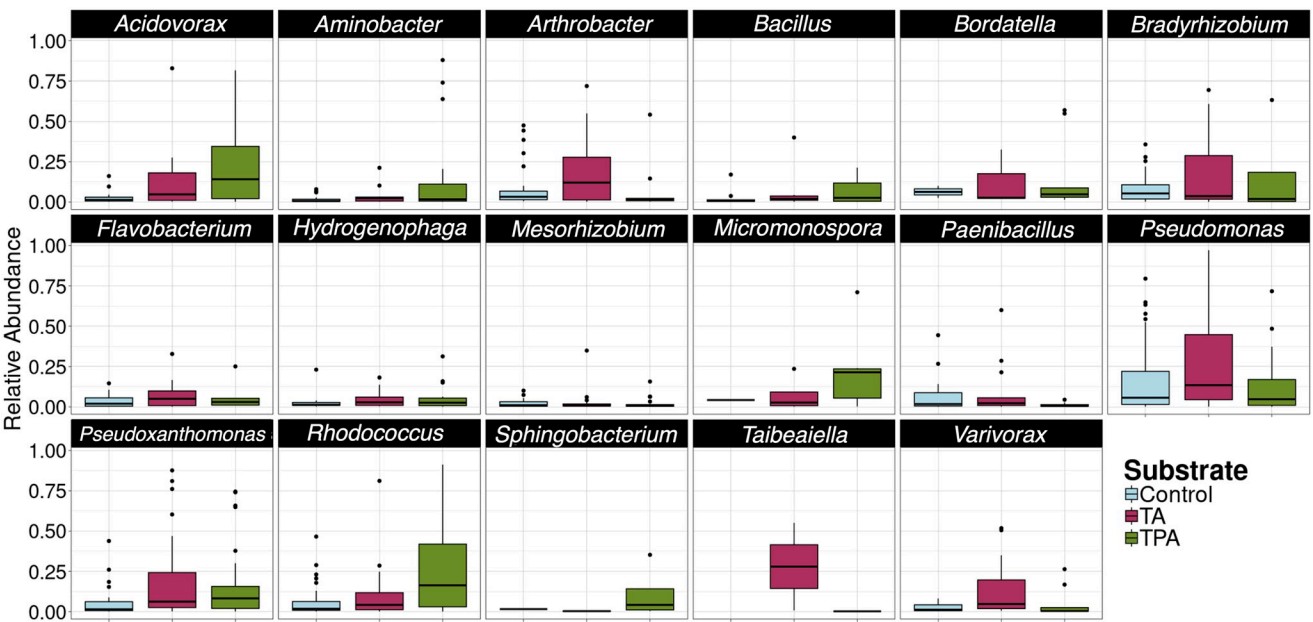

**Fig 4. Select highly abundant genera present at more than 10% relative abundance in at least one sample.** Y-axis shows relative abundance, box and whisker plots show distribution of data. Only data from the fourth transfer is shown. Abbreviations: terephthalamide (TA), terephthalate (TPA).

terephthalamide treatments, numbers represent mean relative abundance in the terephthalate treatment plus or minus the standard deviation (Fig 4). In the samples grown on terephthalamide, *Arthrobacter* (16% ± 18%), *Bradyrhizobium* (15% ± 22%), and *Variovorax* (13% ± 15%) were highly abundant, numbers represent mean relative abundance in the terephthalamide treatment plus or minus the standard deviation (Fig 4). Interestingly, taxa from *Paenibacillus* and *Pseudomonas* had similar relative abundances in the controls, terephthalate, and terephthalamide treatments (Fig 4). Similar trends were observed when samples from all four transfers were shown together, except that *Micromonospora* was much more abundant when grown on terephthalate (S5 Fig in S1 File).

Using the data from all four transfers, a Kruskal-Wallis test was used to identify statistically significant differences in relative abundance between substrate types; five genera had a statistically significant difference in relative abundance between substrates (*Rhodococcus*, *Acidovorax*, *Pseudoxanthomonas*, *Variovorax*, and *Paenibacillus*) (S6 Table in S1 File). A post-hoc test was performed to identify statistically significant differences between pairwise comparisons of substrates (S7 Table in S1 File). *Rhodococcus* and *Paenibacillus* were significantly more abundant in the terephthalate treatment relative to the controls and terephthalamide samples. *Acidovorax* and *Pseudoxanthomonas* were significantly more abundant in the terephthalate and terephthalamide treatments than in the controls. *Variovorax* was statistically more abundant in the terephthalamide treatment compared to the control and terephthalate treatments.

## Unique taxa were enriched in terephthalate treatments compared to terephthalamide treatments

Differential abundance analysis was performed on the data set to look for differentially abundant organisms that were enriched on either terephthalamide or terephthalate relative to the other samples in the data set. Two analyses were performed to identify which taxa are enriched in either the terephthalate or terephthalamide cultures. Since some taxa are higher abundance

in the control relative to either treatment, the terephthalamide samples were compared to all other samples in the dataset (controls and terephthalate samples); similarly, the second analysis compared terephthalate to all other samples in the dataset (controls and terephthalamide samples). In the terephthalamide samples, 47 differentially abundant ASVs were identified; likewise, in the terephthalate samples 111 differentially abundant ASVs were identified. The differentially abundant ASVs are described further in S1 File. To focus on the most abundant enriched taxa, the list of differentially abundant ASVs was filtered to only include those belonging to genera which had relative abundance of 10% or greater and the filtered list was used for further analysis. The most abundant differentially enriched ASVs belonged to five classes: *Actinobacteria* (4 terephthalate enriched, 1 terephthalamide enriched), *Alphaproteobacteria* (19 terephthalate enriched, 6 terephthalamide enriched), *Bacilli* (5 terephthalate enriched, no terephthalamide enriched), *Bacteroidia* (2 terephthalate enriched, 4 terephthalamide enriched), and *Gammaproteobacteria* (15 terephthalate enriched, 7 terephthalamide enriched) (Fig 5).

Many of the enriched ASVs were from the most highly abundant genera (Figs 4 and 5). *Pseudomonas* ASVs were highly abundant in all three treatments, and five *Pseudomonas* ASVs were found to be significantly differentially abundant. Three of the five differentially abundant *Pseudomonas* strains were enriched in the terephthalate treatments, two were annotated as *Pseudomonas mendocina*, *Pseudomonas anguilliseptica* and the third was annotated as a member of the genus *Pseudomonas*. Interestingly, different Pseudomonas ASVs were differentially abundant in the terephthalamide and control treatments (*Pseudomonas frederiksbergensis*, and *Pseudomonas yamanorum*, respectively).

## Prevalence of highly abundant genera across environments

Looking at the prevalence of different genera that were enriched across the five environments, three substrate treatments, and the four transfers shows that some taxa were highly abundant across several environments (Fig 6). For example, *Rhodococcus* was highly abundant in the terephthalate enrichments across all the environments except for the stream sediment enrichments, which were instead dominated by *Acidovorax*. Interestingly, in the second replicate of the terephthalate enrichment inoculated with the Lake Superior sediment, *Aminobacter* dominated instead of *Rhodococcus* in all transfers. The terephthalamide treatments were less consistently dominated by a single taxonomic group and showed more variation between environments. The compost enrichments were dominated by *Arthrobacter* and *Bradyrhizobium*, the hydrocarbon seep enrichments were dominated by *Pseudomonas*, the shoreline sediment enrichments were dominated by *Pseudoxanthomonas*, and the Lake Superior sediment and stream sediment enrichments consisted of many of the same genera enriched in the other environments. The control samples overall had low abundances of the genera that were highly abundant in the other treatments, although *Pseudomonas* and *Arthrobacter* made up a large portion of the communities enriched from the Caspian Sea hydrocarbon seep and the Lake Superior sediment. Many of the samples with high abundances of known contaminant-degrading microbial taxa were also predicted to encode genes for terepthalate biodegradation, protocatechuate aromatic ring cleavage, and amidases (previously implicated in biodegradation of amides) (S6 Fig in S1 File) [27, 48].

## Degradation of terephthalamide and terephthalate

To determine whether terephthalamide and terephthalate were biodegraded during the experiment, high performance liquid chromatography was used to quantify the substrate remaining in one replicate from each environment and transfer at the end of each 14-day growth phase.

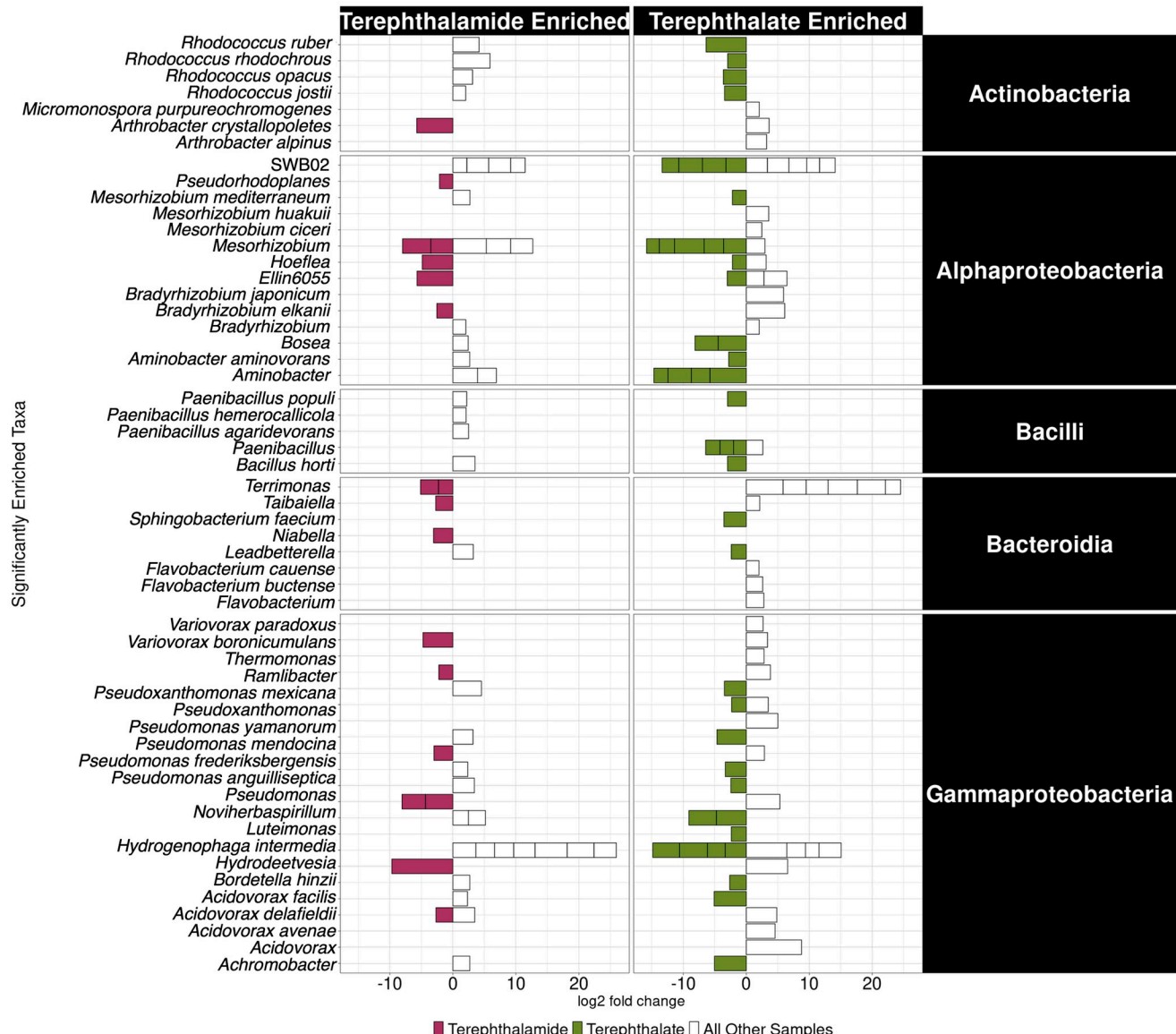

**Fig 5. Differential abundance analysis showed that 47 ASVs were enriched on terephthalamide and 111 ASVs were enriched on terephthalate, relative to the rest of the data set.** Only genera which were present at >10% relative abundance in at least one sample are shown. Genera with multiple ASVs identified as differentially abundant are shown as separate boxes.

After the 14-day growth period, remaining terephthalate was below detection limits in all environments except the Hydrocarbon Seep where 71% was degraded (Figs 7 and 8, S8 Table in S1 File). Results showed that all enrichments showed some degradation of terephthalamide; the average amount of terephthalamide degraded across cultures enriched from all environments was 69.2% (Figs 7 and 8, S8 Table in S1 File).

## Discussion

Selective enrichment is an appealing way to generate microbial communities for biotechnological applications, and for generating simplified communities to study ecological processes in a

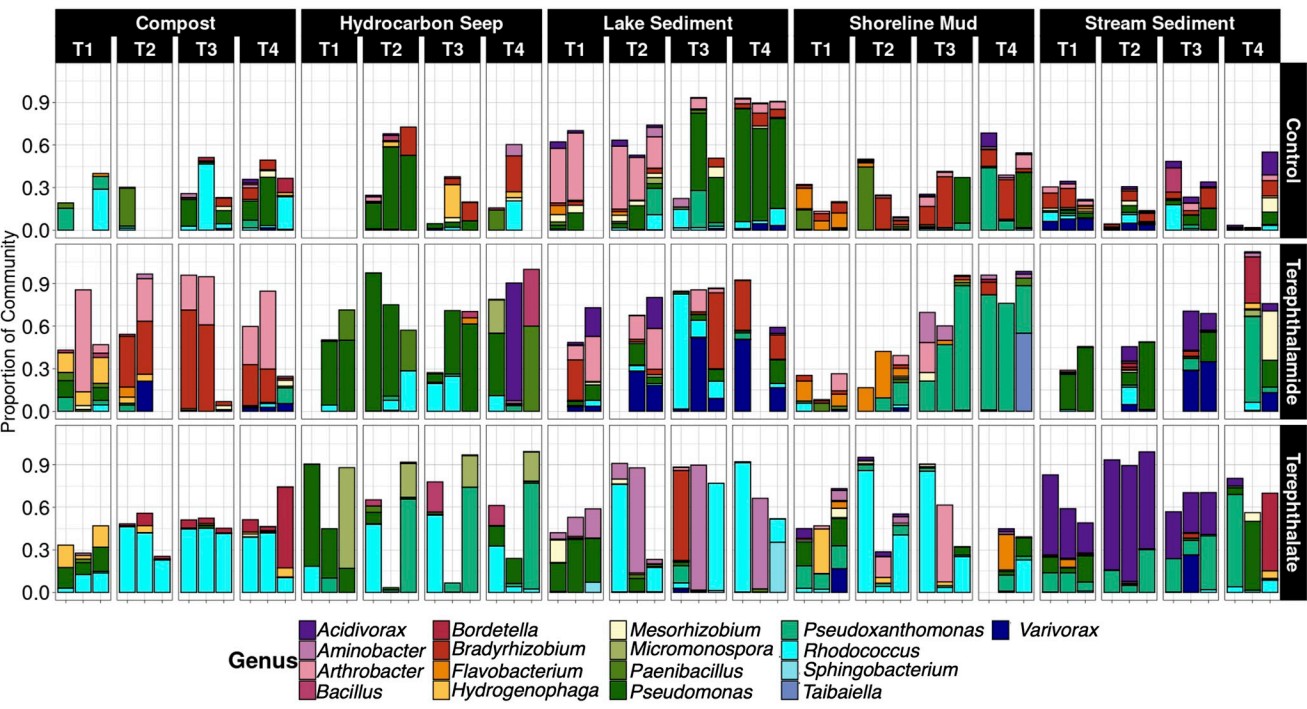

**Fig 6. Relative abundance of key genera across environments and transfers by substrate.**

simplified system [49–51]. Using enriched microbial communities from natural sources may expedite the discovery of new enzymes and pathways that may improve biotechnological applications and understanding of microbial metabolic processes. Since most microorganisms are auxotrophs that rely on byproducts from other cells for growth, community enrichment may

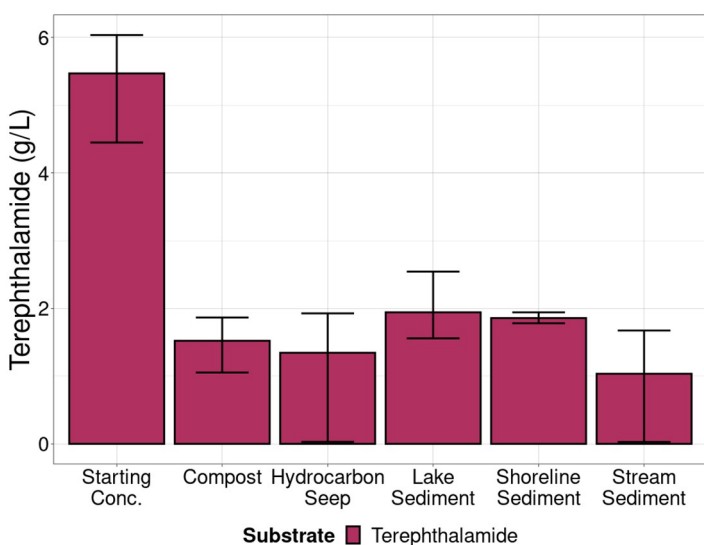

**Fig 7. High performance liquid chromatography quantification of the remaining terephthalamide in one sample from each environment and transfer.** Starting concentration was 5 g/L. Measured starting concentration is represented by the first column on the left. Each bar represents the mean of one replicate from each of the four transfers in each environment (n = 4) or the starting concentration (n = 3).

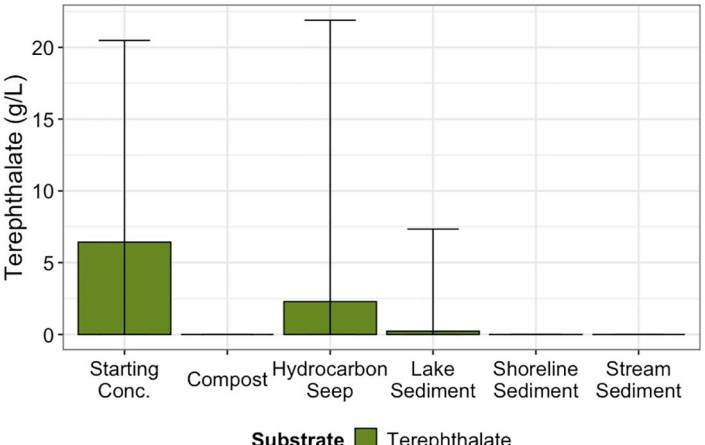

**Fig 8. High performance liquid chromatography quantification of the remaining terephthalate in one sample from each environment and transfer.** Starting concentration was 10 g/L. Measured starting concentration is represented by the first column on the left. Each bar represents the mean of the three replicates in each of four transfers in each environment (n = 12) or the starting concentration (n = 3).

allow us to culture more diverse microorganisms than by using isolation techniques [52]. Here, microbial consortia were enriched to grow on terephthalate and terephthalamide as the sole carbon source. Another study by our group showed increased optical density of similarly enriched communities during one week of growth on plastic-derived substrates [11]. Terephthalate and terephthalamide are aromatic compounds produced by depolymerization of PET through hydrolysis and aminolysis. Organisms able to degrade terephthalate and terephthalamide were enriched from widespread environments: compost, river and stream sediments, saline, and freshwater lake sediments, as well as freshwater shoreline sediment. Despite large error bars in Figs 7 and 8, terepthalate and terephthalamide were degraded in sediment samples from all environments. The variable performance is likely a result of heterogeneity microbial community composition in sediment samples. This demonstrates that the ability of microorganisms to degrade terephthalamide and terephthalate are present in geographically and environmentally diverse environments. Very few previous studies have explored the potential for terephthalamide to be biologically degraded. These enrichments demonstrate clear removal of terephthalamide (Fig 7). While previous work has explored the potential for terephthalate biodegradation to our knowledge this is one of the first studies to demonstrate terephthalamide biodegradation.

A great deal of previous literature has explored terephthalate-degrading organisms and metabolisms because it is foundational to many polyethylene terephthalate degradation, recycling, and upcycling techniques [3, 8–11, 15, 53, 54]. Many of the terephthalate enriched organisms in this study are already well-known terephthalate degraders. Differential abundance analysis showed that four *Rhodococcus* strains were differentially abundant in the terephthalate treatments compared to the controls and terephthalamide. *Rhodococcus* spp. are some of the most well-studied terephthalate degraders [25, 55], have been previously isolated from soil [56] and mangrove sediments [57], and have been implicated in the degradation of many chemicals and pollutants [58, 59]. To date, most isolated *Rhodococcus* strains have come from environments contaminated with recalcitrant pollutants [59]. A *Rhodococcus ruber* isolate was previously shown to degrade polyethylene [32]. Additionally, a *Rhodococcus rhodochrous* isolate was previously shown to degrade diester plasticizers [60], and a *Rhodococcus*

*opacus* isolate was shown to degrade halogenated benzenes, phenols and benzoates [61]. *Rhodococcus jostii* was previously bioengineered to upcycle polyethylene terephthalate waste into value-added bioproducts [15]. Additional known terephthalate-degrading genera were also identified by the differential abundance analysis including *Hydrogenophaga* [62], *Achromobacter* [63], and *Pseudomonas* [54]. The enriched microbial communities were also predicted to encode known terephthalate biodegradation genes. Here, enriched microbial communities contained known terephthalate-degrading organisms as well as genera not previously known to degrade terephthalate: *Paenibacillus*, *Aminobacter*, and *Mesorhizobium* (Fig 5).

In addition to terephthalate-degrading organisms, several potential terephthalamide-degrading microorganisms were identified; these organisms were enriched in the terephthalamide treatments relative to the terephthalate and control treatments. This is, to our knowledge, the first study that has identified potential terephthalamide degraders. Very little is known about the microbial response to terephthalamide, except that it may have antimicrobial properties, especially when it forms a complex with lanthanum nitrate [64]. Terephthalamide monomers have also been intentionally produced via aminolysis as precursors to antimicrobial polyionenes [20]. In this study, microbial strains from four classes (Actinobacteria, Alphaproteobacteria, Bacilli, Bacteroidia, and Gammaproteobacteria) were found to be significantly differentially enriched in the terephthalamide treatments relative to the terephthalate and control treatments. Across the enriched cultures obtained from six unique environments, four genera were highly abundant in the terephthalamide treatments relative to the terephthalamide and control treatments: *Arthrobacter* (some members shown to be degraders of terephthalate [22] and TNT [65]), *Bradyrhizobium* (some members previously implicated in degradation of chlorophenoxyacetic acids [66, 67] and vanillate [68]), *Pseudoxanthomonas* (some members known to degrade bisphenol A from polycarbonate plastic [69] and diesel oil [70]), and *Variovorax* (some members known degrader of phthalate esters [71], and often found in soil contaminated with amide containing compounds [72]) (Fig 4). Three statistically significant differentially abundant ASVs from these genera were identified in the terephthalamide treatments: *Arthrobacter*, *Bradyrhizobium*, and *Variovorax* (Fig 5), showing that these ASVs were more abundant in the terephthalamide-amended enrichments. It has been previously suggested that these genera have bioremediation and/or biodegradation abilities. For example, a psychrotolerant *Arthrobacter alpinus* strain R3.8 was previously isolated from soil collected on the Antarctic Peninsula; genome analysis showed bioremediation potential for naphthalene and aminopolysaacharides [73]. Additionally, *Variovorax boronicumulans* strain CGMCC 4969 was previously isolated from soil and found to degrade acrylamide, a toxic, nitrogen containing compound using a nitrile hydratase [72]. While the mechanism for biological breakdown of terephthalate is known, due to the paucity of studies on biodegradation of terephthalamide, the mechanism for terephathalamide degradation is not known. The enriched microbial consortia were predicted to have known genes for terephthalate biodegradation as well as central carbon metabolism genes for protocatechuate biodegradation (S6 Fig in S1 File) [27]. Additionally, the consortia were predicted to encode genes for amidases, enzymes previously implicated in the biodegradation of amides [48]. Further work is required to confirm the identity of terephthalamide degraders and better characterize the biochemical mechanism for terephthalamide biodegradation.

To our knowledge, this is the first time that a study has shown how geographically and environmentally widespread terephthalate- and terephthalamide-degrading microorganisms are. This finding is supported by prior research studies showing that polymer-degrading microorganisms are common. Alam *et al*. identified PETases in >90% of marine samples, primarily in organisms from the Order *Pseudomonadales* [74]. Additionally, Šerá *et al*. successfully enriched poly (butylene adipate-co-terephthalate) degraders (*Bacilli* and *Actinomycetes*)

from 30 out of 41 agricultural soil samples from the eastern Czech Republic [75]. Additionally, data presented here identifies potential terephthalamide degraders and demonstrates that some microorganisms increase in abundance when given terephthalamide as a sole carbon source, suggesting that terephthalamide is not antimicrobial to certain bacterial species. While only polyethylene terephthalate-derivative degrading metabolisms were investigated in this study, it is likely that there are many more industrially-relevant microbial metabolisms spread across most natural environments.

## Future directions

Future research should attempt to further investigate microbial biodegradation of terephthalamide. Because terephthalamide has a different chemical structure than terephthalate (amide functional groups instead of carboxylic acid functional groups), it is likely that the enzymatic degradation pathway will involve different enzymes than the terephthalate pathway. However, common central metabolic pathways (catechol or protocatechuate) are likely used for both substrates, as was previously shown for the degradation of other aromatic compounds by *Rhodococcus* [59]. Terephthalate is biodegraded through the terephthalate pathway and the breakdown intermediates are completely mineralized to carbon dioxide through central metabolism pathways [27]. Determining the enzymatic pathways for terephthalamide degradation will be important if terephthalamide is an expected derivative of polyethylene terephthalate in a plastic-upcycling system, for example, a system using aminolysis to depolymerize the plastic.

Because the microorganisms highlighted in this study have been enriched in a microbial community, there are many opportunities for future study. First, the communities could be used as a model system to investigate community dynamics, division of labor, as well as cooperation and competition between species within the communities. Further exploration of specialist and generalist species in these enriched communities has been published [27]. Secondly, individual organisms could be isolated from the communities and studied further with whole genome sequencing and laboratory experiments to characterize the individual abilities of each organism apart from the community. Isolates could be bioengineered to upcycle the plastic-derivative compounds into value-added products and then integrated back into the community, or used on their own to form the basis of a plastic upcycling system. It is possible that some of the microorganisms enriched in these communities may be able to degrade both terephthalate and terephthalamide. By growing the communities enriched in this study on mixed terephthalate and terephthalamide or the reciprocal substrate, it may be possible to identify species capable of degrading both substrates, which could have useful biotechnological applications.

Results shown here demonstrate that microorganisms that grow on terephthalate and terephthalamide as a sole carbon source are present in five unique and geographically distributed environments. Future work could screen additional environments for naturally-occurring microorganisms that degrade both the compounds investigated here as well as additional polymers and recalcitrant pollutants. Widespread natural environments may be a reservoir for many more microbial species with useful enzymes and metabolic pathways that will allow for exciting advances in biotechnology and recycling methods in the future.

## Supporting information

**S1 File. Supplemental figures and tables.**
(DOCX)

## Author Contributions

**Conceptualization:** Laura G. Schaerer, Stephen M. Techtmann.

**Formal analysis:** Laura G. Schaerer, Stephen M. Techtmann.

**Investigation:** Laura G. Schaerer, Sulihat Aloba, Emily Wood, Allison M. Olson, Isabel B. Valencia.

**Resources:** Sulihat Aloba, Rebecca G. Ong.

**Supervision:** Stephen M. Techtmann.

**Writing – original draft:** Laura G. Schaerer.

**Writing – review & editing:** Laura G. Schaerer, Stephen M. Techtmann.

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
