## [Decision Letter · Decision Letter 0]

9 Oct 2024

PONE-D-24-41229Enriched microbial consortia from natural environments reveal core groups of microbial taxa able to degrade terephthalate and terphthalamidePLOS ONE

Dear Dr. Techtmann,

Thank you for submitting your manuscript to PLOS ONE. After careful consideration, we feel that it has merit but does not fully meet PLOS ONE’s publication criteria as it currently stands. Therefore, we invite you to submit a revised version of the manuscript that addresses the points raised during the review process.

We look forward to receiving your revised manuscript.

Kind regards,

Gang Xiao

Academic Editor

PLOS ONE

Journal Requirements:

1. When submitting your revision, we need you to address these additional requirements. Please ensure that your manuscript meets PLOS ONE's style requirements, including those for file naming. The PLOS ONE style templates can be found at https://journals.plos.org/plosone/s/file?id=wjVg/PLOSOne_formatting_sample_main_body.pdf and https://journals.plos.org/plosone/s/file?id=ba62/PLOSOne_formatting_sample_title_authors_affiliations.pdf 2. In your Methods section, please provide additional information regarding the permits you obtained for the work. Please ensure you have included the full name of the authority that approved the field site access and, if no permits were required, a brief statement explaining why. 3. Thank you for stating the following financial disclosure: "This work was supported by the Defense Advanced Research Projects Agency ReSource program cooperative agreement HR00112020033. The views, opinions, and/or findings expressed are those of the author and should not be interpreted as representing the official views or policies of the Department of Defense or the U.S. Government. This work was also supported by the Merck Future Insight Prize." Please state what role the funders took in the study.  If the funders had no role, please state: "The funders had no role in study design, data collection and analysis, decision to publish, or preparation of the manuscript." If this statement is not correct you must amend it as needed. Please include this amended Role of Funder statement in your cover letter; we will change the online submission form on your behalf. 4. Please include captions for your Supporting Information files at the end of your manuscript, and update any in-text citations to match accordingly. Please see our Supporting Information guidelines for more information: http://journals.plos.org/plosone/s/supporting-information.

Reviewers' comments:

Reviewer's Responses to Questions

**Comments to the Author**

1. Is the manuscript technically sound, and do the data support the conclusions?

Reviewer #1: Yes

Reviewer #2: Yes

2. Has the statistical analysis been performed appropriately and rigorously? 

Reviewer #1: Yes

Reviewer #2: Yes

3. Have the authors made all data underlying the findings in their manuscript fully available?

Reviewer #1: Yes

Reviewer #2: Yes

4. Is the manuscript presented in an intelligible fashion and written in standard English?

Reviewer #1: Yes

Reviewer #2: Yes

5. Review Comments to the Author

Reviewer #1: Manuscript Number: PONE-D-24-41229

Title: Enriched microbial consortia from natural environments reveal core groups of microbial taxa able to degrade terephthalate and terphthalamide

The paper explored the enrichment of sediment microbial communities and the ability to metabolize specific compounds in a new environment. The researchers enriched microbial communities by collecting sediments from five different environments and used 16S rRNA serialization to identify dominant strains that might be able to break down tetrahydrophthalic acid or terephthalic acid. The core goal is to compare how metabolites spread to the natural environment between microbial communities under controlled and specific substrate rich culture conditions, and to determine whether terephthalic acid is degradable and its potential degrading microorganisms.

The English of the entire manuscript must be revised.

1. Abstract: There are too many descriptions about the background, and the summary is not prominent enough to be a qualified abstract. In addition, whether to add the conclusion part to make the article more complete and distinguish it from the abstract part, highlighting the novelty and achievement of the article.

2. There are many problems and errors in the language format, please check and correct the full text. In addition, attention should be paid to the format of references and the use of Spaces.

e.g. Line 93; Line 101; Line 136: “10000 x g”; Line 148; Lines 156-157: ‘’; Lines 159-160: -; Line 180: Terephthalamide; Lines 211-212; Line 218, etc.

3. Introduction: There are too many subjective descriptions in the last paragraph of the introduction, such as we, our.

4. Line 178: um

5. Lines 211-212: Missing space “±”

6. Line 180: Check whether the first letter of Terephthalamide needs to be capitalized in full text.

7. Results: The section title does not summarize part of the content well, please modify it.

8. The figures definition are too low and need to be improved.

9. The strain name in the figure also needs to be italicized.

10. Discuss: Lines 364-365: The analysis of the conclusion lacks objectivity.

11. our/we sentences should be changed.

12. As for the lack of data analysis, it was limited to the functional analysis of the existing flora, which did not achieve the role of analyzing and discussing the results. The author should further analyze the results in combination with the literature and experimental content.

Reviewer #2: The authors enriched species are able to metabolize the microbial communities of PET hydrolysis and ammonia decomposition products, and this work is important for the isolation and purification of functional strains that efficiently transform pollutants from PET decomposition products, and to achieve the recovery of PET by biological methods and high value.

6. PLOS authors have the option to publish the peer review history of their article (what does this mean?). If published, this will include your full peer review and any attached files.

Reviewer #1: No

Reviewer #2: No

---

## [Author Response · Author response to Decision Letter 0]

21 Nov 2024

Response to Reviewers – Schaerer et al 

We appreciate the feedback from the editor and the reviewers. We have updated the manuscript to address the reviewers’ comments.

Journal Requirements:

We have indicated the no permits were required for these samples. 

3. Thank you for stating the following financial disclosure: "This work was supported by the Defense Advanced Research Projects Agency ReSource program cooperative agreement HR00112020033. The views, opinions, and/or findings expressed are those of the author and should not be interpreted as representing the official views or policies of the Department of Defense or the U.S. Government. This work was also supported by the Merck Future Insight Prize."

This statement is correct. 

We have included captions for supporting information files at the end of the manuscript.

Reviewer #1

The paper explored the enrichment of sediment microbial communities and the ability to metabolize specific compounds in a new environment. The researchers enriched microbial communities by collecting sediments from five different environments and used 16S rRNA serialization to identify dominant strains that might be able to break down tetrahydrophthalic acid or terephthalic acid. The core goal is to compare how metabolites spread to the natural environment between microbial communities under controlled and specific substrate rich culture conditions, and to determine whether terephthalic acid is degradable and its potential degrading microorganisms.

The English of the entire manuscript must be revised.

We have made substantial revisions to the manuscript to address the reviewers comments

1. Abstract: There are too many descriptions about the background, and the summary is not prominent enough to be a qualified abstract. In addition, whether to add the conclusion part to make the article more complete and distinguish it from the abstract part, highlighting the novelty and achievement of the article.

The background description in the abstract has been reduced and the conclusions have been expanded to highlight the novel scientific contribution of this article.

2. There are many problems and errors in the language format, please check and correct the full text. In addition, attention should be paid to the format of references and the use of Spaces.

e.g. Line 93; Line 101; Line 136: “10000 x g”; Line 148; Lines 156-157: ‘’; Lines 159-160: -; Line 180: Terephthalamide; Lines 211-212; Line 218, etc.

These errors have all been addressed, with the exception of “Kruskal-Wallis” as the correct syntax is hyphenated (https://datatab.net/tutorial/kruskal-wallis-test).

3. Introduction: There are too many subjective descriptions in the last paragraph of the introduction, such as we, our.

Most instances of “our” and “we” have been removed from the text. The statements “to our knowledge…” have been preserved since we do not know of another way to communicate the intent of these phrases.

4. Line 178: um

This has been updated to “μm”

5. Lines 211-212: Missing space “±”

This has been addressed.

6. Line 180: Check whether the first letter of Terephthalamide needs to be capitalized in full text.

This has been resolved.

7. Results: The section title does not summarize part of the content well, please modify it.

The subtitles in the Results section have been checked and re-worded accordingly.

8. The figures definition are too low and need to be improved.

The figures are formatted according to the journal specifications and are at the resolution needed by the journal and have been processed through the PACE tool to ensure compliance with the journal’s specifications.

9. The strain name in the figure also needs to be italicized.

We have changed the figures to italicize genus and species names.

10. Discuss: Lines 364-365: The analysis of the conclusion lacks objectivity. our/we sentences should be changed.

Most instances of “our” and “we” have been removed from the text. The statements “to our knowledge…” have been preserved since there is not another way to communicate the intent of these phrases.

12. As for the lack of data analysis, it was limited to the functional analysis of the existing flora, which did not achieve the role of analyzing and discussing the results. The author should further analyze the results in combination with the literature and experimental content.

We have included further discussion of biodegradation of terephthalate and terephthalamide. Including some discussion of the lack of previous studies on terephthalamide degradation. Additionally, we performed additional analysis with PICRUSt2 to assess the predicted genome content of the enriched samples; this has been included in the supplemental information and discussion has been added to the text.

Reviewer #2

The authors enriched species are able to metabolize the microbial communities of PET hydrolysis and ammonia decomposition products, and this work is important for the isolation and purification of functional strains that efficiently transform pollutants from PET decomposition products, and to achieve the recovery of PET by biological methods and high value.

1. In line 384, the Latin designation of the strain is not italicized.

Line 384 only contains class designations, which do not need to be italicized (https://wwwnc.cdc.gov/eid/page/scientific-nomenclature#:~:text=Italicize%20family%2C%20genus%2C%20species%2C,letter%20but%20are%20not%20italicized). We have italicized genus labels in figures and throughout the paper.

2. The size of the error bars in Figure 8 is much larger than the mean value, does this indicate the real reliability of the data

Enrichment cultures are all descended from unhomogenized soil samples, so some heterogeneity in microbial community composition is expected. Additional justification of this has been added to the text (Discussion section, first paragraph).

3. Whether the enriched terephthalamide-degrading bacteria are a single functional bacterium performing the degradation function or a variety of functional bacteria synergistically involved in the efficient degradation of this pollutant.

It is impossible to definitively determine this from a microbial community without functional (multi-omics) analyses. It is likely that synergistic interactions between bacterial species contribute to biodegradation in these enriched microbial communities. This has been called out as an area of future research (Future Directions section, second paragraph).

4. Is there any change in microbial biomass before and after the degradation of these two pollutants, increase or decrease?

Biomass measurements were outside the scope of the current study. Another study by our research group showed an increase in optical density in similar cultures grown on plastic-derived substrates (Schaerer et al. 2023, Journal of Industrial Microbiology), discussion of this has been added (Discussion section, first paragraph).

5. Possible products of microbial degradation of the two pollutants? Are the two substrates transformed into high value products by microorganisms, or are they completely metabolized to produce CO2, and gaseous nitrogen pollutant emissions?

Terephthalate is metabolized into carbon dioxide and the pathways for terephthalamide are currently unknown (although it is likely metabolized to carbon dioxide as well). Additional clarification has been added (Future Directions section, first paragraph).

---

## [Editor Report · Decision Letter 1]

26 Nov 2024

Enriched microbial consortia from natural environments reveal core groups of microbial taxa able to degrade terephthalate and terphthalamide

PONE-D-24-41229R1

Dear Dr. Techtmann,

We’re pleased to inform you that your manuscript has been judged scientifically suitable for publication and will be formally accepted for publication once it meets all outstanding technical requirements.

Kind regards,

Gang Xiao

Academic Editor

PLOS ONE
---

## [Editor Report · Acceptance letter]

13 Dec 2024

PONE-D-24-41229R1 

PLOS ONE

Dear Dr. Techtmann, 

I'm pleased to inform you that your manuscript has been deemed suitable for publication in PLOS ONE. Congratulations! Your manuscript is now being handed over to our production team.

Kind regards, 

on behalf of

Dr. Gang Xiao 

Academic Editor

PLOS ONE